# Human-AI Alignment of Learning Trajectories in Video Games: a continual RL benchmark proposal

**Yann Harel**[1,2,†]**, Lune Bellec**[1,2,3,†]**, Francois Paugam**[1,2,3]**, Hugo Delhaye**[1,2]**, Audrey Durand**[3,4,5]

`{yann.harel,lune.bellec,francois.paugam,hugo.delhaye}@umontreal.ca,`
`audrey.durand@ift.ulaval.ca`

[1]**Centre de Recherche de l'Institut Universitaire de Gériatrie de Montréal**
[2]**Psychology Department, Université de Montréal**
[3]**Mila**
[4]**Computer and Software Engineering Department, Université Laval**
[5]**Canada CIFAR AI Chair**

[†] Equal contribution

## Abstract

We propose a design for a continual reinforcement learning (CRL) benchmark called GHAIA, centered on human-AI alignment of learning trajectories in structured video game environments. Using *Super Mario Bros.* as a case study, gameplay is decomposed into short, annotated scenes organized into diverse task sequences based on gameplay patterns and difficulty. Evaluation protocols measure both plasticity and stability, with flexible revisit and pacing schedules. A key innovation is the inclusion of high-resolution human gameplay data collected under controlled conditions, enabling direct comparison of human and agent learning. In addition to adapting classical CRL metrics like forgetting and backward transfer, we introduce semantic transfer metrics capturing learning over groups of scenes sharing similar game patterns. We demonstrate the feasibility of our approach on human and agent data, and discuss key aspects of the first release for community input.

## 1 Introduction

There is increasing interest in continual reinforcement learning (CRL) scenarios, where even the best agents can continue improving indefinitely in their environment (Abel et al., 2023). Effective CRL is an important stepping stone towards artificial general intelligence (AGI), with the potential to match or exceed human-level adaptability. However, despite humans representing a *de facto* gold standard of adaptive intelligence, CRL benchmarks rarely establish where human learning trajectories lie. We propose here a video Game Human-AI Alignment (GHAIA) benchmark grounded in extensive, high-quality longitudinal human data. Video game environments can be used to train RL agents (Nichol et al., 2018), are amenable to controlled manipulation (Delfosse et al., 2024), and are engaging enough to collect large amounts of human gameplay (Sharma et al., 2024). Using a staple environment for RL research as a case study (the *Super Mario Bros.* videogame (Nintendo, 1985)), we outline a GHAIA benchmark with two distinguishing features:

1. **Annotated elementary tasks:** The videogame is decomposed into hundreds of short, semantically annotated tasks (scenes), which can be procedurally rearranged into different task sequences. This large combinatorial space allows precise control over task transitions—gradual or abrupt—and difficulty progression, enabling the study of transfer, forgetting, and stability in various conditions.

2. **Human learning trajectories:** Extensive behavioural and physiological data were collected from human players undergoing a controlled sequence of tasks comparable to that of a CRL agent. This enables not only comparison to final human performance, but also direct alignment with human learning trajectories across the task sequence.

After a short review of prior art on CRL benchmarks, we outline the general design of the GHAIA benchmark. We then present the human dataset on *Super Mario Bros.*, and describe how the game can be decomposed into scenes to characterize human learning trajectories. We also show how human trajectories can be contrasted with that of a traditional RL agent to provide measures of human alignment. As we aim for a first release of the GHAIA benchmark and associated competition in 2026, the discussion section outlines several directions of short-term and long-term development. Community input will be critical to shape the benchmark to ensure it is relevant to current research priorities and impactful for the future of human-aligned, lifelong-learning AI.

## 2 Prior art

Video games have long been a staple of RL research. A pioneering effort, based on an open-source version of *Super Mario Bros.* (Nintendo, 1985), built a RL benchmark by adding an API for reinforcement learning and the generation of procedural levels to the game (Karakovskiy & Togelius, 2012). Retro games have been further leveraged in the Arcade Learning Environment (ALE) (Bellemare et al., 2013), offering a large collection of distinct games. However, these collections were not designed with cross-task transfer in mind: mastering one game typically provides little advantage in another. Even in the Gym Retro competition (Nichol et al., 2018), where levels from the same franchise (e.g., *Sonic the Hedgehog*; SEGA, 1991) were used, transferring knowledge proved challenging. This limited potential for generalization makes these benchmarks suboptimal to study continual learning. While rich environments like Minecraft have demonstrated potential for transfer, the complex nature of the game makes it difficult to characterize what aspects of task similarity promote or hinder reuse (Tessler et al., 2017).

More recent efforts have shifted toward environments specifically designed to enable controlled studies of continual learning under environment shifts, often starting with an existing videogame RL environment and adding an explicit manipulation of certain characteristics of the game. For example, CRLMaze (Lomonaco et al., 2020) used a 3D game engine to create visually distinct maze tasks organized into task sequences. HackAtari (Delfosse et al., 2024) introduced both visual and physical modifications to classic Atari games through memory manipulation. While these benchmarks can help understand learning under perceptual change, it is unclear how they manage to capture continual learning of new *behaviour*, which is a key aspect of RL.

The Continual Doom (COOM) benchmark (Tomilin et al., 2023) partly addresses this gap, by generating a series of tasks inspired by the game *Doom* (id Software, 1993), and proposing a series of task sequences designed to systematically test not only shift in the perception of the game, but also in terms of game objectives as well as difficulty. The COOM benchmark still investigated a limited number of tasks and task sequences. Importantly, none of the continual RL benchmarks reviewed here included human baselines.

Recent work has emphasized simply optimizing a reward may not lead to human-aligned behavior and fall short of human levels of adaptability and generalizability (Sharma et al., 2024), which are highly desirable qualities for CRL. In order to measure human-AI alignment, Sharma et al. (2024) proposed a series of manually annotated tasks and situations, with a clear taxonomy of behaviors, within a complex multi-player game. Although promising, this environment is not amenable to controlled task sequence manipulations which are ideal for a CRL benchmark, and human behaviour was also collected outside any controlled sequence, making it impossible to compare human and AI *learning trajectories*. Our proposed GHAIA benchmark aims to fill this gap by modifying a classic game environment to support a vast space of interpretable task sequences, grounded in meaningful human learning trajectories as baselines.

# 3 GHAIA benchmark design for CRL

## 3.1 Scene and level segmentation

The core design idea behind the GHAIA benchmark for CRL is to treat a series of levels in a retro videogame such as *Super Mario Bros.* (Nintendo, 1985) as a task sequence. We propose to segment all levels into short scenes, each associated with detailed annotations of game patterns. For example, valid annotations would be the presence of groups of enemies, gaps, or stairs. A given AI agent would be trained sequentially on each level independently, possibly circling through this sequence multiple times, with a set number of attempts per scene and per level. After being trained on a given level, the agent would be evaluated on previous and future tasks to assess behavioral metrics such as forgetting and transfer. Scene segmentation would allow for the creation of new level compositions and corresponding task sequences. Moreover, detailed pattern annotations would provide fine control on the characteristics of a given task sequence along a number of dimensions.

## 3.2 Procedural Task Sequence Generation

To design diverse and principled task sequences, we implement a procedural generation procedure based on inter-scene pattern distances. Let $n$ denote the number of scenes that have been obtained after segmenting the levels of a videogame. Each of the $n$ scenes is annotated with a binary vector encoding the presence of specific gameplay patterns considered. From this annotation, we define a pairwise distance matrix between scenes using the Jaccard distance over their pattern vectors. We define a task sequence as a partition of the $n$ scenes into $m$ disjoint pseudo-levels $\mathcal{S} = \{\mathcal{T}_1, \ldots, \mathcal{T}_m\}$, each $\mathcal{T}_i$ comprising about $n/m$ scenes, and presented in a fixed order. The number of possible such task sequences $\mathcal{S}$ is immense, due to combinatorial explosion, enabling exploration of highly diverse task sequences. To construct a task sequence procedurally, we define objective functions over the set of candidate partitions. These objectives include:

- **Smoothness**: minimizing the average pattern distance between successive pseudo-levels and within each pseudo-level;

- **Contrast**: maximizing inter-level dissimilarity to increase transitions between gameplay patterns;

- **Coverage**: maximizing the diversity of patterns observed across the entire curriculum;

- **Difficulty Pacing**: imposing a monotonic increase in average scene difficulty across pseudo-levels, using human performance as a proxy.

We can implement greedy and stochastic generation procedures to sample task sequences under various regimes for each objective (or combination of objectives). This procedural framework lays the foundation for benchmark variants that target specific combination of learning challenges, including stability under high shifts in game patterns or incremental generalization across similar patterns of varying difficulty.

## 3.3 AI training and performance metrics

To evaluate CRL in a GHAIA benchmark, we adopt standard metrics from the CL literature, specifically (Wołczyk et al., 2021), tailored to the structure of short annotated scenes. These metrics aim to assess not only retention and forgetting, but also adaptation and learning dynamics over scene curricula. The agent is trained in successive steps $i$ on each of the task sequence $(\mathcal{T}_i)_{i=1}^m$, with a fixed number of training steps $\Delta$, uniformly assigned to the $n/m$ scenes included in the task set. Note that for a given scene, the agent spawns in one of the available human savestates available at that scene.

The **performance** of an AI agent after finishing training on the $j$-th task is evaluated as the success rate on a task $i$: $p_i(j) \in [0, 1]$. The overall performance $P(j)$ denotes the average success rate across all tasks (including $j$). The overall performance after training on the last task $(\mathcal{T}_m)$, $P(m)$, serves as the main metric for tuning hyper-parameters and comparing overall effectiveness.

We measure **forgetting** $F_i$ in AI agent behaviour on task $\mathcal{T}_i$ as the drop in performance between the end of its training and the end of the curriculum $p_i(i) - p_i(m) \in [-1, 1]$, and the overall forgetting $F$ is the average over all tasks. Lower values indicate better memory retention. A **forward** transfer metric can also be derived by comparing training curves of a reference, naive agent, with the agent having completed all tasks, see (Woł czyk et al., 2021) for details. If the task sequence is repeated more than once, backward transfer can also be tracked.

It should be noted that all CRL performance metrics presented here rely on evaluating agents **without training**. This is impossible with humans: if an individual is asked to perform a task to evaluate its performance, learning inevitably occurs. We now present how to administer a sequence of tasks to humans in a controlled way that mimics a CRL benchmark, in order to extract proxy measures of established CRL metrics.

# 4   Super Mario Bros.: A case study

We apply the proposed GHAIA design to generate a CRL benchmark using *Super Mario Bros.* (Nintendo, 1985), an iconic 2D side-scrolling platformer released on Famicom and Nintendo Entertainment System (NES). We implement the game using `gym-retro` (Nichol et al., 2018), a Python library created to facilitate RL agent design and experiments within retro videogames environments. This library features more than 10 console emulators, which allows the emulation of thousands games of various genres. *Super Mario Bros.* consists of discrete levels structured around a set of recurring gameplay mechanics, including running, jumping, avoiding or defeating enemies, and navigating obstacles such as gaps, pipes, and moving platforms. Despite its apparent simplicity, the game presents a rich variety of micro-challenges that demand fine motor coordination, temporal precision, and adaptive planning. Its level design has previously been studied in both game design and AI research, making it an ideal candidate for systematic analysis of learning dynamics in artificial agents and humans alike. We extend this design by decomposing the game into a library of short, self-contained tasks, annotate them structurally, and analyze how both humans and agents learn to solve them.

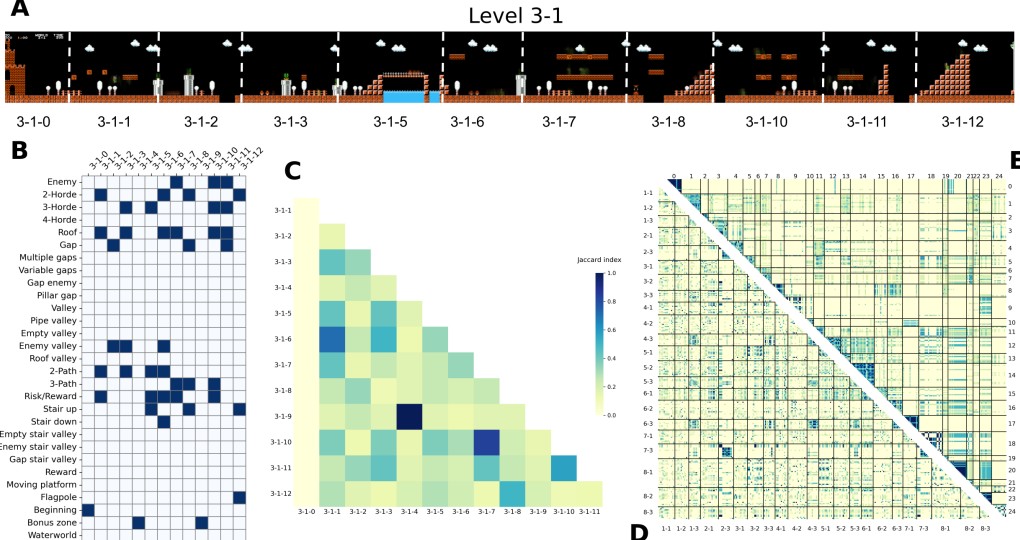

Figure 1: **Scene annotations.** (A) Visual layout of level 3-1 split into 13 short scenes (minus bonus scenes 4 and 9). (B) One-hot encoding of annotated game patterns per scene. (C) Jaccard distance matrix between pattern encoding of level 3-1 scenes. (D) Extended Jaccard distance matrix across all 313 scenes of 22 levels in their original order, and (E) ordered in 25 pattern-driven clusters.

### 4.1 Scene segmentation

To enable fine-grained analysis of learning dynamics, we segmented a subset $m = 22$ (see Supplementary Material for a list) of the 32 original levels of *Super Mario Bros.* into $s = 313$ elementary gameplay segments, hereafter referred to as scenes. Each scene captured a short, self-contained challenge (mean duration $\approx 5$ s) and was designed to isolate a distinct problem-solving context.

We defined scene boundaries manually, following level design principles from Smith et al. (2008). Segmentation points were placed at transitions between semantically distinct gameplay elements, e.g. after an enemy cluster or before a gap jump, with neutral traversal zones serving as natural dividers, see Figure 1A. This ensured that each scene posed a specific micro-task, such as timing a jump or navigating a staircase. Each scene was indexed by entry and exit coordinates derived from emulator RAM, allowing consistent segmentation across human and agent gameplay.

### 4.2 Structural annotation of scenes

We annotated each scene using a structured taxonomy of 28 gameplay design patterns, adapted from Dahlskog & Togelius (2012) and expanded to better capture the variety of challenges in *Super Mario Bros.* These patterns describe recurring design elements such as enemy clusters, gaps of various widths, multi-path layouts. All scene metadata, including pattern annotations and emulator memory boundaries, is available in a publicly released CSV file[1]. This file enables automatic extraction of gameplay in specific scenes for both humans and AI agents.

Figure 1B presents the binary annotation matrix for level 3-1, showing the presence or absence of each pattern in the 13 scenes. This annotation scheme provides a semantic fingerprint for each scene and supports a variety of downstream tasks, including curriculum design, difficulty estimation, and interpretable performance evaluation. The Jaccard distance matrix between the annotations of the scene shows a clear pattern of repetition, both within and across levels (Figure 1C-D). A hierarchical clustering was applied to reorder the whole matrix into 25 clusters of scenes, shown in Figure 1E. These clusters provide an interpretable low-dimensional organization of the scene space, supporting procedurally generated task sequences with highly homogeneous scene composition and sharp transitions across levels.

### 4.3 Human training

We use human gameplay data collected as part of the Courtois Neuromod Project (Boyle et al., 2020). Five participants played *Super Mario Bros.* while undergoing simultaneous behavioral, brain and physiological recording. Participants completed between 13 and 18 hours of gameplay (totaling 84 hours), using a custom MRI-compatible gamepad replicating the layout of a classic SNES controller (Harel et al., 2022). The videogame was presented via `gym-retro` (Nichol et al., 2018), enabling real-time logging of player inputs and a posteriori reconstruction of internal game state variables (e.g., position, score, lives) via the emulator's RAM. A detailed description of the dataset and acquisition parameters can be found in CNeuroMod (2025)

The participants were exposed to the original levels design without interruptions between scenes. Players were exposed to each level in order and given as many attempts (each including up to three lives) as necessary in order to complete the level once, before starting to practice the next level. So instead of a fixed amount of gameplay steps $\Delta$ provided to AI agents, human players were subject to a form of early stopping. This sequence of task presentations, called here *discovery phase*, largely mirrors the proposed CRL task sequences for a particular choice of levels. A notable difference is that scenes situated at the end of a difficult level are typically repeated less than scenes at the beginning. Once subjects had completed all $m = 22$ levels, a new *practice* phase begun, where levels were presented in pseudo-random fashion (ensuring all levels were presented the same number of times), mirroring traditional i.i.d. training in RL.

---

[1] will become available after the anonymous review period

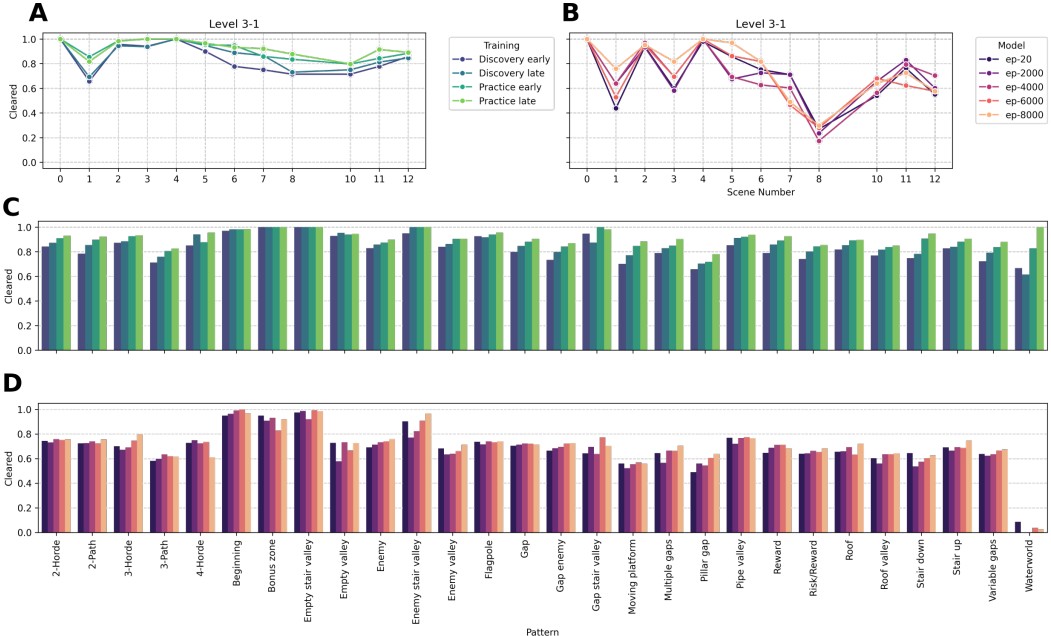

Figure 2: **Human vs AI learning trajectories.** Scene-level success rate across learning stages for level 3-1 in humans (A) and PPO agent (B). (C-D) Same as A-B, but for scenes grouped into patterns.

## 4.4   Human vs AI trajectories

To illustrate the comparison of AI and human learning, we trained a baseline agent using a traditional i.i.d. RL and Proximal Policy Optimization (PPO) (Schulman et al., 2017) across 20 levels (see Supplementary Material for details). Although the agent training does not follow the proposed GHAIA RCL task sequences, we can still study how AI performance evolved throughout training steps. The agent architecture consisted of a four-layer convolutional neural network with an actor-critic head, receiving grayscale frame stacks as input (resolution: $84 \times 84$; stack size: 4). The action space corresponded to discrete button presses in `gym-retro`.

Training comprised 8000 PPO episodes, each consisting of 512 rollout steps per level followed by policy updates. Rollouts were initialized by replaying a random number of actions from a human `.bk2` trajectory (sub-01), ensuring in-distribution but non-trivial starting states. One successful reference trajectory was used per level. Each rollout got re-initialized after a game over or a level completion. The reward function combined positive feedback for forward movement and in-game score increase, with penalties for life loss and time spent. Policy checkpoints were saved at episodes 20, 2000, 4000, 6000, and 8000 in order to assess learning trajectories.

To compare human and AI learning trajectories, we further divided the gameplay of the *discovery* and *practice* phases of human training into early and late, through median split of attempts. Note that the late *discovery* measures can be used to generate a proxy of $p_i(i)$ for humans, and the early *practice* phase to generate a proxy of $P(m)$, although human learning continues to occur during these phases as well. The average success rate is presented (averaged between subjects) across these phases of training in Figure 2A, for $i$ equal to level 3-1. High variability in the difficulty of scenes is evident, with scenes 1 and 6-10 in particular in the early discovery phase. Clear learning effects are also present on nearly all scenes, with notably a jump of nearly 20% in success rate for scene 1. By contrast, the PPO agent (Figure 2B) had markedly lower performance than humans on some scenes, e.g. scene 8. While some scenes exhibited learning patterns analogous to humans, such as

scene 1, others showed no learning effects (e.g. scene 8). Even worse, scene 11 for example showed a negative impact of training on performance, suggesting interference between tasks.

Finally, Figure 2C and D summarize learning progress by design pattern over the full dataset, revealing distinct acquisition profiles for humans and artificial agents, with humans displaying consistent improvements across patterns. These metrics can capture the transfer of learning across coherent semantic groups that span multiple scenes and levels.

# 5 Discussion

This paper proposes a design for a GHAIA benchmark for CRL, using *Super Mario Bros.* as a case study, and seeking input of the community. We have in particular not yet benchmarked CRL models to establish what kind of computational budget is required in order to achieve meaningful performance, both in terms of model type, model size and amounts of training data per level. We discuss below several other potential use cases and extensions that may be prioritized for the first release of the benchmark and competition.

## 5.1 Curriculum learning

The large combinatorial space of scenes naturally lends itself to the procedural generation of task sequences. This capability can be used in multiple ways. One option is to create a small number of canonical task sequences varying in the abruptness of difficulty and gameplay pattern transitions, to serve as standardized tracks in the benchmark. Another option is to generate random task sequences as a basis for evaluation. A third possibility is to pose curriculum optimization itself as a learning challenge—where the goal is to discover sequences that maximize the CRL performance of a given agent. A complementary direction is to analyze which structural properties make a task sequence more or less difficult, potentially under constraints (e.g., limited number of game patterns per sequence). While all these avenues are viable, a concrete choice will need to be made for the first benchmark release and competition.

## 5.2 Quantitative human-AI alignment metrics

In this work, we presented the evolution of performance metrics $p_i(i)$ for human players. Because our PPO agent was trained using standard i.i.d. reinforcement learning, direct comparisons between human and agent metrics were not possible. However, we showed that introducing *ad hoc* training phases for the agent enabled meaningful interpretation.

When training CRL agents on the canonical Mario level sequence, it becomes feasible to directly compare $p_i(i)$ between humans and artificial agents. This comparison comes with caveats. For example, AI agents are typically constrained by a fixed training budget $\Delta$, whereas humans are trained with early stopping. Humans are also evaluated on a single task sequence, while AI will be exposed to different sequences. Beyond these differences, we would ideally compare $P(m)$ across agents and humans—but unlike agents, humans cannot be evaluated under "frozen weights." As a workaround, the early *practice* phase of human gameplay may serve as a proxy for such evaluation, as well as for estimating forgetting.

Additionally, by contrasting the learning curves observed during the *practice* and *discovery* phases in human gameplay, it may be possible to define a formal analogue of backward transfer. The only classic CRL metric that appears fundamentally unapproximable with our current dataset is forward transfer, as it is not feasible to collect data from a single individual performing simultaneously all tasks in sequence or independently for the first time. However, we can define new measures of progress at the level of game patterns, thus establishing metrics that are sensitive to the *semantic* transfer of knowledge across tasks.

### 5.3 Expanding beyond the case study

Our team has also collected data on two additional games: *Super Mario All-Stars* (a 16-bit remake with altered visuals and physics) and *Super Mario Bros. 3*, which includes a combination of overlapping and novel game patterns compared with *Super Mario Bros.* (Thompson, 2015). Importantly, the same participants who played the original *Super Mario Bros.* game also completed these games under controlled conditions, enabling cross-game analysis of learning trajectories. These datasets offer the potential to study generalization under more dramatic domain shifts, such as changes in visual style and gameplay structure. A set of standardized annotations for game patterns that extend across the Mario franchise has already been proposed. Creating scene cuts (with corresponding annotations) and integrating these games into the benchmark will require significant effort, but could become a priority if supported by the community.

### 5.4 Expanding measures of human alignment with brain and physiological activity

While our case study focused on the behavioural component in *Super Mario Bros.*, this human sample was designed first for applications in cognitive neuroscience, and also features a broad array of physiological signals collected concurrently with gameplay. Specifically, brain activity was recorded using functional magnetic resonance imaging (fMRI), which provides an indirect measure of neural activation at high spatial but limited temporal resolution. Additional modalities include eye tracking (attention and motor preparation), respiration, cardiac activity, and skin conductance (sympathetic and parasympathetic indicators of arousal and emotion). A separate Mario-EEG dataset captures neural activity at high temporal resolution, and another dataset is being collected using magnetoencephalography (MEG), which offers a middle ground between EEG and fMRI.

Taken together, these signals offer a multidimensional view of human cognitive and affective state, and make the mario human dataset the most detailed of its kind to date. This array of measurements could be used to assess whether the internal representations of an RL agent align with distinct human-derived latent spaces. It has been suggested that aligning latent representations with brain activity may improve robustness and generalization in AI systems (Freteault et al., 2025). These multimodal signals provide a promising complement to behavioural imitation and offer a more holistic measure of human-likeness. However, incorporating them into the benchmark would require substantial engineering and may delay an initial competition.

### 5.5 Limitations

One clear limitation is that only five individuals currently serve as human baselines for CRL. Capturing human learning under tightly controlled conditions is resource-intensive, especially with physiological data. Given the difficulty of collecting longitudinal highly controlled data, substantially expanding the participant pool would likely require a coordinated, multi-lab effort, even for behavioural data alone.

Another limitation is the total number of scenes (313). While this allows for a vast number of possible sequences, many such permutations may differ only trivially. Some agents may eventually saturate the benchmark. Nonetheless, the fact that human participants are still refining their performance suggests nontrivial complexity remains. Moreover, integration of brain and physiological data would increase alignment difficulty and mitigate the risk of trivial saturation. Expansion to other games, as described above, also offers a clear path to increased benchmark complexity.

### 5.6 Broader Impact Statement

CRL has the potential to shape the development of AI agents capable of lifelong adaptation, and this benchmark offers opportunities for cross-disciplinary inquiry. For instance, if optimal task sequences emerge for AI agents, this may transfer to humans and may reveal shared principles of effective learning curricula which can be applied to video games aimed at teaching skills over time.

**Acknowledgments**

The Courtois project on neural modelling (Julie A. Boyle, Basile Pinsard, Amal Boukhdhir, Sylvie Belleville, Simona Brambatti, Jeni Chen, Julien Cohen-Adad, Andre Cyr, Adrian Fuente, Pierre Rainville, and Pierre Bellec, 2020) was made possible by a generous donation from the Courtois foundation, administered by the Fondation Institut Gériatrie Montréal at CIUSSS du Centre-Sud-de-l'île-de-Montréal and University of Montreal. The Courtois NeuroMod team is based at "Centre de Recherche de l'Institut Universitaire de Gériatrie de Montréal", with several other institutions involved. See the cneuromod documentation for an up-to-date list of contributors (https://docs.cneuromod.ca).

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

# Supplementary Materials

*The following content was not necessarily subject to peer review.*

## 6   Taxonomy of game patterns in Super Mario Bros

| Category | Pattern | Definition |
|---|---|---|
| *Enemies* | | |
| | Enemy | Single enemy. |
| | 2-Horde | Two adjacent enemies. |
| | 3-Horde | Three adjacent enemies. |
| | 4-Horde | Four adjacent enemies. |
| | Roof | Enemy below low ceiling. |
| *Gaps* | | |
| | Gap | One terrain gap. |
| | Multiple gaps | Series of equal gaps. |
| | Variable gaps | Gaps of differing width. |
| | Gap enemy | Enemy flying above gap. |
| | Pillar gap | Pipe/block stands between gaps. |
| *Valleys* | | |
| | Valley | A valley surrounded by vertical obstacles, no Piranha pipes. |
| | Pipe valley | Valley with Piranha pipes. |
| | Empty valley | Valley without enemies. |
| | Enemy valley | Valley with enemies. |
| | Roof valley | Valley with overhead ceiling. |
| *Multiple paths* | | |
| | 2-Path | One platform, two routes. |
| | 3-Path | Two platforms, three routes. |
| | Risk/Reward | Hazardous path offering reward. |
| *Stairs* | | |
| | Stair up | Ascending block steps. |
| | Stair down | Descending block steps. |
| | Empty stair valley | Valley between stairs, no enemies. |
| | Enemy stair valley | Valley between stairs with enemies. |
| | Gap stair valley | Valley between stairs with floor gap. |
| *Additions to Dahlskog & Togelius (2012)* | | |
| | Reward | Isolated coin/power-up. |
| | Moving platform | Horizontally or vertically mobile platform. |
| | Flagpole | Level-end goal pole. |
| | Beginning | Initial segment after start. |
| | Bonus zone | Hidden area rich in collectibles. |

Table 1: Taxonomy of *Super Mario Bros.* design patterns used to annotate scenes, adapted from Dahlskog & Togelius (2012).

## 7   Selection of game levels in Super Mario Bros

We selected a subset of 22 levels composed of levels 1–1, 1–2, 1–3, 2–1, 2–3, 3–1, 3–2, 3–3, 4–1, 4–2, 4–3, 5–1, 5–2, 5–3, 6–1, 6–2, 6–3, 7–1, 7–3, 8–1, 8–2, and 8–3. We excluded boss levels (*-4)

and underwater levels (2–2, 7–2), as these involve distinct game mechanics and design patterns not addressed in the current taxonomy (Dahlskog & Togelius, 2012).

# 8 Difficulty of game patterns for humans

To examine broad structural trends, we aggregated performance by design pattern (Figure 3A) and by scene cluster (Figure 3B), where clusters group scenes with similar pattern compositions. Our results reveal stable difficulty gradients across structural categories and gameplay motifs, providing a fine-grained behavioral reference for evaluating adaptation, difficulty pacing, and alignment in agent-based continual learning. This structural framework serves as the foundation for analyzing *Super Mario Bros.* gameplay at the scene level.

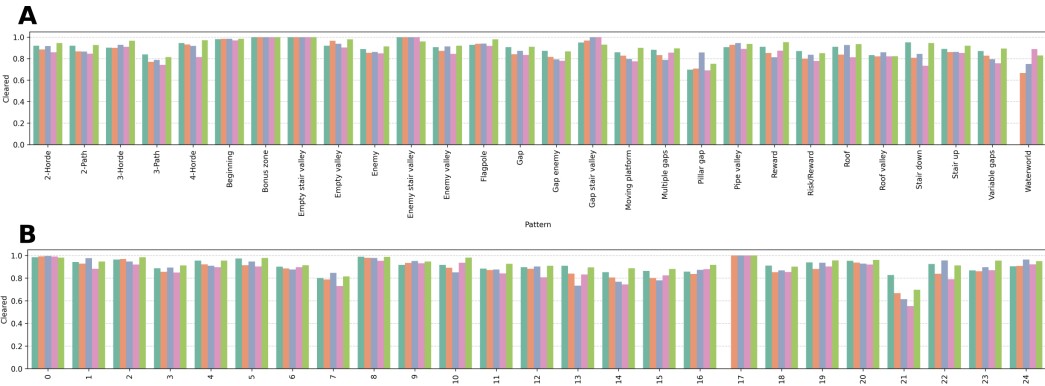

Figure 3: **Human gameplay performance across scenes, patterns, and clusters.** (A) Average success rate per design pattern across participants, showing consistent differences in pattern difficulty. (B) Average success rate per cluster (scene grouping based on pattern structure), highlighting cluster-level variation in difficulty across players.

# 9 Scene selection for training the PPO agent

The agent was trained on 20 of the 22 levels included in the benchmark. Levels 5–1 and 6–3 were held out for evaluation to assess generalization, and are not used in this work. While level 5–1 primarily differs in visual style (except from level 7-1), level 6–3 introduces both aesthetic and mechanical variations, including narrower platforms and altered player traction, posing a more challenging domain shift.