# OpenReview forum: "Human-AI Alignment of Learning Trajectories in Video Games: a continual RL benchmark proposal"
_rl-conference.cc/RLC/2025/Workshop/RLVG — RLVG Workshop - RLC 2025_

### Official Review · Reviewer_JEWU · 2025-06-15
**Interesting benchmark but incomplete analysis**

**Rating:** 3
**Confidence:** 4

**Summary:**

The paper proposes a new benchmark environment for Continual Reinforcement Learning (CRL) in video games called Game Human-AI Alignment (GHAIA). As the name suggests, the benchmark also proposes recorded human trajectories to study the alignment between a trained AI agent and human gameplay. The framework is based on the Super Mario Bros. (SMB) video game. GAHIA has two main features: annotated elementary tasks, which define hundreds of short scenes taken from the game, with each scene annotated with some game features that describe the particular scene (e.g. number of enemies, number of jumps, etc…); human learning trajectories: human trajectories recorded while the players learn how to play the game.

The authors train a Proximal Policy Optimization (PPO) agent on a set of levels and compare the learning performance of the agent with that of humans. What is interesting is that the authors do not compare only the pure performance at the end of each level, but rather how the agent and the humans *learn* how to solve a level after learning the previous one. Results show a similar trend between humans and agents; however, I believe the results are not enough to support the claims the authors make throughout the paper. More details in the next sections.

**Strengths:**

I think the benchmark is really interesting and a good fit for the workshop. I believe the problem is important but understudied, and I think we need more papers and research studying CRL in games—not only for a path to AGI (as the authors say) but also for applied RL in games (e.g., adapting a game agent to what the players do). Although the benchmark is not complete, I think the way the authors define and construct it is interesting and could be useful.

**Weaknesses:**

As I mentioned in the previous sections, the flow of the paper is not optimal. Even if the authors claim that the paper is not complete and (I think) the goal is to submit it to the workshop to get suggestions and feedback, the paper lacks many details, discussions, and baselines.

**Best Paper Nomination:**

No

**Claims:**

The problem and benchmark that the authors propose are very interesting and a good fit for the workshop. However, I believe the paper is very confused in the flow and the claims are not well supported by the experiments. As the authors say, this is a work in progress, and they submitted this paper to ask for support from game AI community to improve the benchmark. However, I think the paper needs a little bit more work to be accepted.

1. In my opinion, the paper is not easy to read and the flow is very confusing. First, the authors frame the benchmark as a general CRL benchmark, and treat SMB as a case study, while the benchmark and features definition are very tied to that particular game and I do not know how the same set of features could be applied to a different game. Then, the paper lacks descriptions of many details (more on this at point 5).
2. I am not sure comparing “human learning trajectories” with what the agent does is completely fair. SMB is a very popular game, and many people already know its main gameplay dynamics and mechanics. Instead, an RL agent needs to learn everything from scratch. I would suggest that the authors expand more on this.
3. Curriculum learning is never defined in the paper until 5.1 (that is basically at the end of the paper). Moreover, the authors say “the large combinatorial space of scenes naturally lends itself to the procedural generation of task sequences,” but as far as I understood, this is not tested, because the authors test the agent only on 20 levels.  Combining scenes can be not trivial, so I would suggest that the authors expand on this.
4. I believe the comparison with the trained agents is not complete, since the authors use a simple PPO baseline that is notoriously known to be not efficient for CRL tasks. A fairer comparison would be with a CRL algorithm. Similarly, the Prior Art section does not touch any previous CRL work, especially in games. I think this is a significant lack in the discussion. Some references: Loss of plasticity in deep continual learning, Dohare et al., 2024; Overcoming catastrophic forgetting in neural networks, Kirkpatricka et al., 2017; Policy fusion for adaptive and customizable reinforcement learning agents, Sestini et al., 2022.
5. Some definitions and discussions are just mentioned, while I would suggest the authors to expand more on these: the metrics (smoothness, contrast, etc..), how the scenes are constructed, and how the scenes the be procedurally combined together are just mentioned, while a proper description is needed (e.g. mathematical notation, examples, etc..). This is important for someone who wants to use the benchmark. It is unclear how the physiological signal can be used to study CRL in agents (”This array of measurements could be used to assess whether the internal representations of an RL agent align with distinct human-derived latent spaces.”).

**Suggestions:**

*Provide any other feedback or suggestions for improvement.*

For a detailed discussion with examples, I suggest that the authors read the “Claims” section. In brief:

1. Improve the flow of the paper, adding details to the concepts introduced by the authors;
2. Expand the discussion on the comparison between human- and agent-learning trajectories;
3. Add more CRL baselines to the discussion; and
4. Expand the Prior Art section with CRL references and discussions.

---

### Official Review · Reviewer_jMSG · 2025-06-16
**Human-AI Alignment of Learning Trajectories in  Video Games: a continual RL benchmark proposal**

**Rating:** 3
**Confidence:** 4

**Summary:**

In the article, the authors propose the creation of a benchmark for continual reinforcement learning called GHAIA (Game Human-AI Alignment). Using the Super Mario Bros NES game, the authors propose to decompose the game into multiple scenes that then could be categorized based on patterns (game objects in the game) or difficulty (compared with human baselines. The proposed benchmark will also offer human baselines, that allow to compare agents and humans in continual learning metrics like forgetting and backward transfer.

**Strengths:**

The paper appears to be well written
Directly comparing learned behaviour between humans and machines is an interesting contribution to the community
Several interesting discussions that help define and show the possibilities of the final benchmark

**Weaknesses:**

The analysis does not use the proposed benchmark
+ "the agent training does not follow the proposed GHAIA RCL task sequences"
+ Thus, I find the analysis not comparable between agents and humans, as it is discussed in section 5.2.

Only data for 5 individuals

**Best Paper Nomination:**

No

**Claims:**

Yes. Since this article gives an outline of the proposed benchmark, I feel that not much evidence is necessary.
The analysis also shows some of the gathered human data in the game.

**Suggestions:**

Although not from the area of human alignment, I saw no description of how to compare the alignment of an RL agent with humans. From my understanding, the concept of human alignment looks at how the learned behaviour is aligned with the behaviour of humans, namely the policy. Without further descriptions, at least for me, it is confusing to say measures of human alignment and then compare something like the learning behaviour and not what is actually learned.

I don't understand why the rollouts are initialized with a random number of actions from the human trajectories. Why is this essential?

"introducing ad hoc training phases for the agent enabled meaningful interpretation", I don't understand why

---

### Decision · Program_Chairs · 2025-06-19

**Decision:**

Accept

**Comment:**

This paper introduces **GHAIA**, a new benchmark for continual reinforcement learning in video games, using Super Mario Bros. to compare human and AI learning trajectories. It proposes categorizing game scenes and collecting human baselines for robust comparative analysis.

The paper is well-written and the idea of directly comparing human and AI learning behaviors is an innovative contribution to the field. The discussions effectively outline the benchmark's potential, addressing an important yet understudied area in game AI and continual RL.

However, the current analysis does not properly utilize the GHAIA benchmark, and the agent training does not adhere to the proposed task sequences, which hinders meaningful comparisons. The paper flow could also be improved with more detailed descriptions for key components, and could use more suitable Continual Reinforcement Learning (CRL) algorithms as baselines instead of simply PPO, with more discussion of prior CRL work. We encourage the authors to address these points in the camera-ready version for presentation at the workshop.